# A simple method to estimate flow restriction for dual ventilation of dissimilar patients: The BathRC model

Andrew R. Plummer[1], Jonathan L. du Bois[1], Joseph M. Flynn[1], Jens Roesner[1], Siu Man Lee[2], Patrick Magee[3], Malcolm Thornton[2], Andrew Padkin[2], Harinderjit S. Gill[1,4]*

**1** Department of Mechanical Engineering, University of Bath, Bath, North Somerset, United Kingdom, **2** Royal United Hospitals NHS Foundation Trust, Bath, North Somerset, United Kingdom, **3** BMI Bath Clinic, Bath, North Somerset, United Kingdom, **4** Centre for Therapeutic Innovation, University of Bath, Bath, North Somerset, United Kingdom

* r.gill@bath.ac.uk

## Abstract

### Background

With large numbers of COVID-19 patients requiring mechanical ventilation and ventilators possibly being in short supply, in extremis two patients may have to share one ventilator. Careful matching of patient ventilation requirements is necessary. However, good matching is difficult to achieve as lung characteristics can have a wide range and may vary over time. Adding flow restriction to the flow path between ventilator and patient gives the opportunity to control the airway pressure and hence flow and volume individually for each patient. This study aimed to create and validate a simple model for calculating required flow restriction.

### Methods and findings

We created a simple linear resistance-compliance model, termed the BathRC model, of the ventilator tubing system and lung allowing direct calculation of the relationships between pressures, volumes, and required flow restriction. Experimental measurements were made for parameter determination and validation using a clinical ventilator connected to two test lungs. For validation, differing amounts of restriction were introduced into the ventilator circuit. The BathRC model was able to predict tidal lung volumes with a mean error of 4% (min:1.2%, max:9.3%).

### Conclusion

We present a simple model validated model that can be used to estimate required flow restriction for dual patient ventilation. The BathRC model is freely available; this tool is provided to demonstrate that flow restriction can be readily estimated.

**Models** and data are available at DOI 10.15125/BATH-00816.

**Data Availability Statement:** All data files are available from the University of Bath Research Data Archive. DOI 10.15125/BATH-00816.

**Funding:** One of the authors, PM, was employed as an anaesthetist by a commercial organisation (BMI Bath Clinic) at the time the work was performed. This funder provided support in the form of salary for author PM and providing access to the facilities for conducting experiments, but did not have any additional role in the study design, data collection and analysis, decision to publish, or preparation of the manuscript. The specific roles of these authors are articulated in the 'author contributions' section.

**Competing interests:** No conflict of interest. The support from BMI Bath Clinic does not alter our adherence to PLOS ONE policies on sharing data and materials.

## Introduction

The current COVID-19 crisis could risk ventilator capacity shortfall. If necessary, ventilator capacity could be increased by ventilating two patients using the machine, namely Dual Patient Ventilation (DPV). This would require two circle systems, connected in parallel, as per the work of Neyman & Irvin [1], and Paladina et al. [2]. Complexity arises when patients have different airway and lung impedances as each would require different ventilator settings. Hence, distributing and controlling pressure and flow to each patient, independently, is a significant challenge. Accordingly, the Anesthetic Patient Safety Foundation (APSF) has mandated against ventilator sharing [3].

The COVID-19 crisis has renewed interest in DPV and, in March 2020, this approach was introduced in New York, adopting the "Columbia Protocol" of ventilation [4]. This relies on careful matching of patient characteristics, and its effectiveness would be sensitive to changes in patient compliance. A new experimental study by Tronstad et al. [5] in relation to COVID-19 concluded that large discrepancies were found in delivered tidal volumes for paired test lungs with compliance differences. Furthermore, high Positive End-Expiratory Pressure (PEEP) could strongly influence the distribution of tidal volume. They were unable to reliably overcome this hazard.

Evidence is emerging that lung compliance in COVID-19 patients is not as reduced as in other forms of Acute Respiratory Distress Syndrome, ARDS [6]. Nevertheless, DPV for patients with differing characteristics, particularly tidal volume due to differing compliances, will still require a modified breathing circle. This refers to the introduction of an impedance (resistance or compliance) to appropriately distribute the supplied tidal volume. Increasing the resistance in the inspiratory limb of the patient with either the higher lung compliance or the one requiring a lower tidal volume seems plausible. This is the subject of this paper. A team from Hospital Geel, Belgium, have been experimenting with the same technique [7], and there has been a recent simulation study [8]. This latter work succinctly reviews previous work on DPV and the challenges involved.

A single circle system for one patient has a unidirectional valve in each of its inspiratory and expiratory limbs. These are usually integrated into the anaesthetic machine circle system attachments. A key risk of DPV is inadvertent sharing of gas flows either between patients or between the inspiratory and expiratory limbs of a single patient. Installing two parallel circle systems, each with two directional control valves, restores some control over this risk. This arrangement reduces dead space for each half of the system, and the potential for $CO_2$ rebreathing. The testing reported here does not include studies of how the arrangement manages $CO_2$.

Mathematical modelling and simulation of both human respiratory and mechanical ventilation systems is invaluable to help understand novel scenarios such as DPV. Characterising lung mechanical properties using resistance and compliance has become commonplace. Estimated values are available from studies such as Arnal et al. [9], although other modelling approaches are possible as reviewed in Carvalho and Zin [10]. Complete system models have also been developed as reported in Wilson et al. [11], and these have been used to extensively study low flow breathing systems [12]. However, the aim of our study was to determine if a very simple analytical model can adequately predict behaviour of a ventilator system. An analytical model permits direct calculation of the flow restrictor resistance required to achieve a specified tidal volume.

## Methods

### A linear lumped resistance-compliance (RC) network model: The BathRC model

A highly simplified lumped resistance-compliance model (Fig 1) can be used to represent single patient ventilation with four terms: linear resistance ($R_v$) and compliance ($C_v$) for the

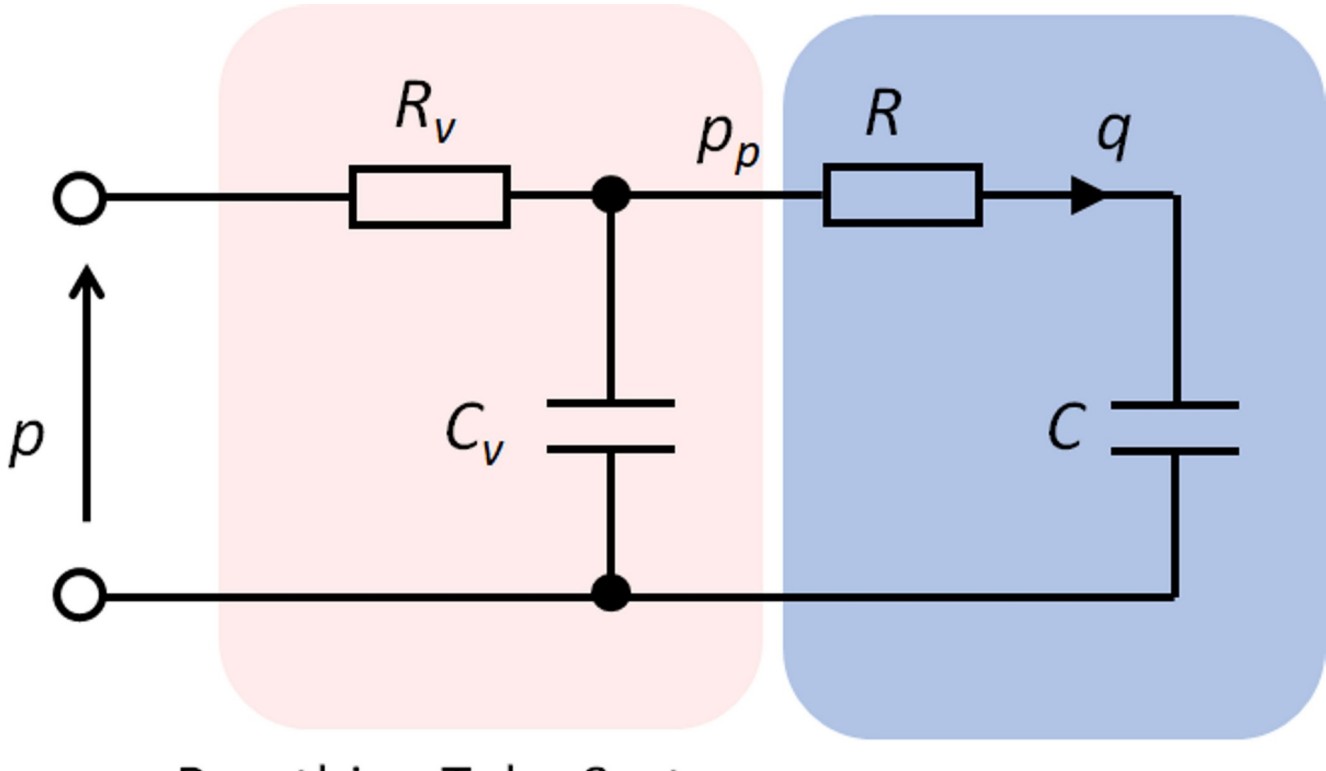

**Fig 1. Simple RC network model of ventilator system and patient, with linear resistance ($R_v$) and compliance ($C_v$) for the ventilator tubing system, and linear resistance ($R$) and compliance ($C$) for the patient.** The RC model calculates tidal volume as a function of variations in patient $R$ and $C$ values. It also gives the relationships between airway flow rate, $q$, airway pressure, $p_p$, and ventilator pressure, $p$.

ventilator tubing system, and linear resistance ($R$) and compliance ($C$) for the patient. The inspiration and expiration phases can be represented with different parameters, resulting in up to eight parameters per patient. This RC model calculates tidal volume as a function of variations in patient $R$ and $C$ values. It also gives the relationships between airway flow rate, $q$, airway pressure, $p_p$, and ventilator pressure, $p$ (full equations in S1 File). A DPV-specific use case is to calculate the required inspiration restriction ($R_r$) to operate with an increased ventilator pressure (necessitated by ventilator sharing) without increasing the patient's tidal volume undesirably. Experimental measurements are required for model parameter estimation and to validate the model. The experiments are described below.

## Experimental measurements

Experiments were performed using an Aisys CS$^2$ (Software version 8.0, GE Healthcare, Chicago, USA) anaesthetic ventilator operated in Pressure Control Mode. This is the most appropriate setting for DPV, as the settings for a single patient would not need to change for ventilating two identical patients. In this mode the adjustable settings are:

- Pinsp (the inspiration pressure in excess of PEEP).

- PEEP (Positive end-expiratory pressure), the ventilator pressure during expiration.

- RR (the respiratory rate, breaths per minute).

- I:E (inspiration to expiration time ratio).

In Pressure Control Mode, the ventilator effectively controls the driving pressure to transition between PEEP and Pinsp+PEEP as quickly as possible at the required switching times; any limits or triggers which might alter this profile need to be disabled for dual patient use.

The ventilator was connected to a Silverknight 22 mm circle system (Intersurgical Ltd, Wokingham, UK) in conjunction with Heat and Moisture Exchange (HME) filters (Clear-Therm 3, Intersurgical Ltd). Two fixed test lungs (Test Lung 190, Siemens Healthineers, Erlangen, Germany) were used for the experiments, these were termed *Lung 1* and *Lung 2*. Two Fluke VT Plus HF Gas Flow Analysers (Fluke Biomedical, Everett, Washington, USA) were used to make flow measurements, and data were collected by connecting each analyser to a personal computer (Dell XPS13 i5, Dell UK, Bracknell, UK) running Vent Tester for Windows software (version 2.01.07, Fluke Biomedical). Data were collected at 50 Hz on each personal computer. Custom functions (MATLAB 2019b, The Mathworks Inc., Natick, MA, USA) were used to co-register the data collected on the two computers for each experiment.

Fig 2 details three configurations that were used for experimental validation of the model. The characterisation experiments used single circuit configurations (Fig 2A, Circuit 1, and 2b, Circuit 2) and validation measurements used dual circuit configurations (Fig 2C, Circuit 3). The dual circuit layout contained four sets of non-return valves (Ref: 1950000, Intersurgical Ltd), also known as one-way valves to prevent sharing of gas flows and to handle expired $CO_2$ adequately; the proposed circuit (Fig 2C) used additional non-return valves in each inspiratory and expiratory limb to stop inspiration or expiration back flows and to reduce each system's dead space.

## Parameter estimation

Circuit 1 was used to estimate flow resistance parameters for different components. Two types of restrictor were tested, the first was a non-return valve and the second was a novel flow restrictor with a very small orifice (11.7 mm$^2$ effective cross-section), hereafter termed *small orifice restrictor* or *SOR*. The SOR device was 3D printed (Form 2, Formlabs Inc., Somerville, MA, USA) for testing purposes. A minimum of 15 cycles of data were collected at each of three different Pinsp pressures, 5, 15, 25 cmH$_2$O; for all tests PEEP was set to 5 cmH$_2$O, RR was 15 breaths/min, and I:E ratio was 1:2. The use of the two flow meters in Circuit 1 were used to measure the pressure drop ($\Delta p$) across each tested component. For each flow restrictor, all pressure drop measurements were plotted against the mean flowrate ($Q_m$), the average of the flowrates measured from the two flowmeters. A bi-square weighted robust least squares fitting method (Matlab 2019b, The MathWorks, Natick, MA, USA; the bi-square method was used for outlier rejection in the experimental data) was used to fit the quadratic function given in equation 1,

$$\Delta p \;=\; K_2 Q_m{}^2 + K_1 Q_m + K_0 \tag{1}$$

The quadratic function represents a combination of turbulent and laminar losses in a flow. This equation becomes linear by setting $K_2 = 0$, and proportional by additionally setting $K_0 = 0$. Hence, the differences between quadratic, linear and constrained (proportional) linear fits were examined. With a proportional fit, $K_1$ is the resistance value of the component being tested. The BathRC model formulation can only accommodate the proportional linear representation of a restrictor.

Circuit 2 was used to estimate the compliance and resistance values for both test lungs (each lung was tested separately). Here, L1 = L2 = 0.4 m to give a total tube length of 2.32 m

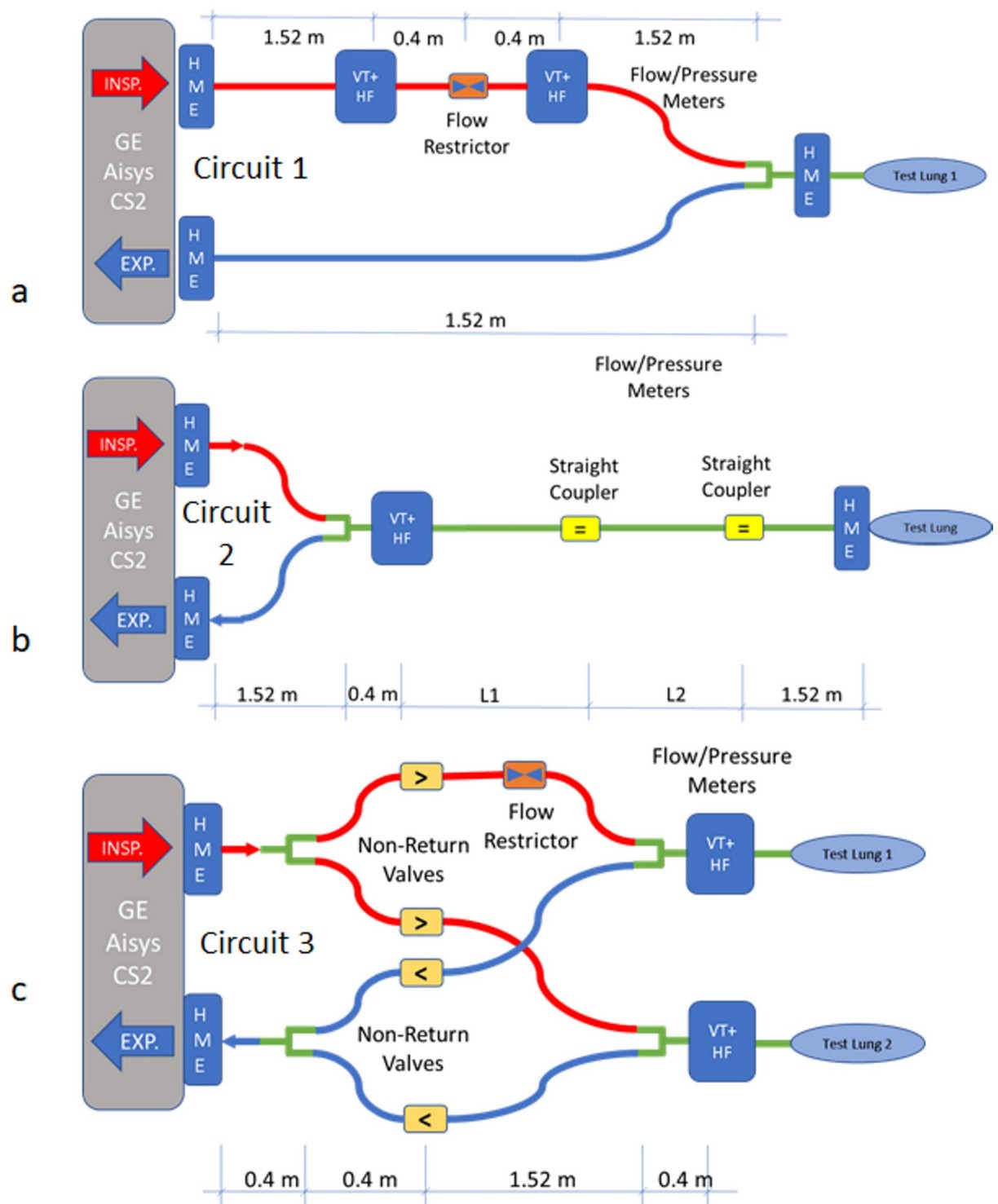

**Fig 2. Ventilators circuits used for testing and validation.** HME = heat and moisture exchanger, VT+ HF = Fluke VT+ HF flow meter, INSP. = inspiration, EXP. = expiration.

beyond the flow meter. The compliance of the tubing itself was estimated using L1 = L2 = 1.52 m, to make the total length 4.56 m, and replacing the test lung with a blockage at the end of the tube.

For all three tests, at least 15 cycles were again collected at each of three different Pinsp pressures, 5, 15, 25 cmH$_2$O, keeping the values of PEEP, RR and I:E fixed as above.

## Validation of BathRC model

Circuit 3 was used for validation and data were collected for the Pinsp pressures of 15 and 25 cmH$_2$O (other ventilator settings were fixed). The following five validation tests were conducted with different flow restrictors connected in the inspiratory limb for Lung 1 (Fig 2C):

1. Pinsp = 25 cmH$_2$O and no restrictor present (replaced by straight connector)

2. Pinsp = 25 cmH$_2$O and non-return valve acting as restrictor

3. Pinsp = 25 cmH$_2$O and SOR device

4. Pinsp = 15 cmH$_2$O and no restrictor present (replaced by straight connector)

5. Pinsp = 15 cmH$_2$O and non-return valve acting as restrictor

The measured values of tidal volume (obtained from the integration of the measured flows) were compared predictions given by the BathRC model and estimated parameters (as above).

## Results

### Parameter estimation

Fig 3A shows the two linear fits for pressure drop versus mean flowrate (Circuit 1 tests) in the case of the non-return valve. The corresponding fit data is presented in Table 1, with R$^2$ values of 0.99 and 0.75 for linear and proportional fits, respectively. The data show that this non-return valve has a cracking pressure of approximately 1 cmH$_2$O. For the SOR, the quadratic and constrained linear fits are shown in Fig 3B. The fits (Table 1) result in R$^2$ values of 0.99 and 0.89 for quadratic and proportional fits, respectively. Resistances for the two devices were estimated from the gradient of the proportional fits. The non-return valve resistance, $R_r$, was 12 cmH2O/L/s, and the SOR resistance was 33 cmH2O/L/s.

For parameter estimation (Circuit 2), $R$ and $C$ values for Lung 1 were 12 cmH$_2$O/(L/s) and 0.040 L/cmH$_2$O, respectively. For Lung 2, $R$ and $C$ values were 10 cmH$_2$O/(L/s) and 0.030 L/cmH$_2$O, respectively. The ventilator tubing resistance ($R_v$) and compliance ($C_v$) values were 22 cmH$_2$O/(L/s) and 0.004 L/cmH$_2$O respectively.

## Validation of BathRC model

For validation test 1 (Fig 4A, top row), without any additional resistance in the inspiratory limb for Lung 1, the flow was quite similar between the two test lungs. However, the higher compliance of Lung 1 was reflected in higher peak flow values. For both test lungs the BathRC model predictions for lung volume change were similar to the measured data (Fig 4A, middle and bottom rows).

For validation test 2, adding the non-return valve to the inspiratory limb for Lung 1 caused the peak flows for Lung 1 to be lower than those for Lung 2 (Fig 4B, top row). Consequently, the tidal volume was reduced for Lung 1 (Fig 4B, middle row), with model predictions again closely tracking the measured data (Fig 4B, middle and bottom rows). The tidal volume for Lung 1 was reduced by 22% relative to its unrestricted state.

In validation test 3, restriction in the form of the SOR device further reduced the peak flow values for Lung 1 (Fig 4C, top row). For Lung 1, the tidal volume reduction was 36% from its unrestricted state. For Tests 1 to 3, there was no change in the parameters for Lung 2, so the model predicted the same tidal volume in each case.

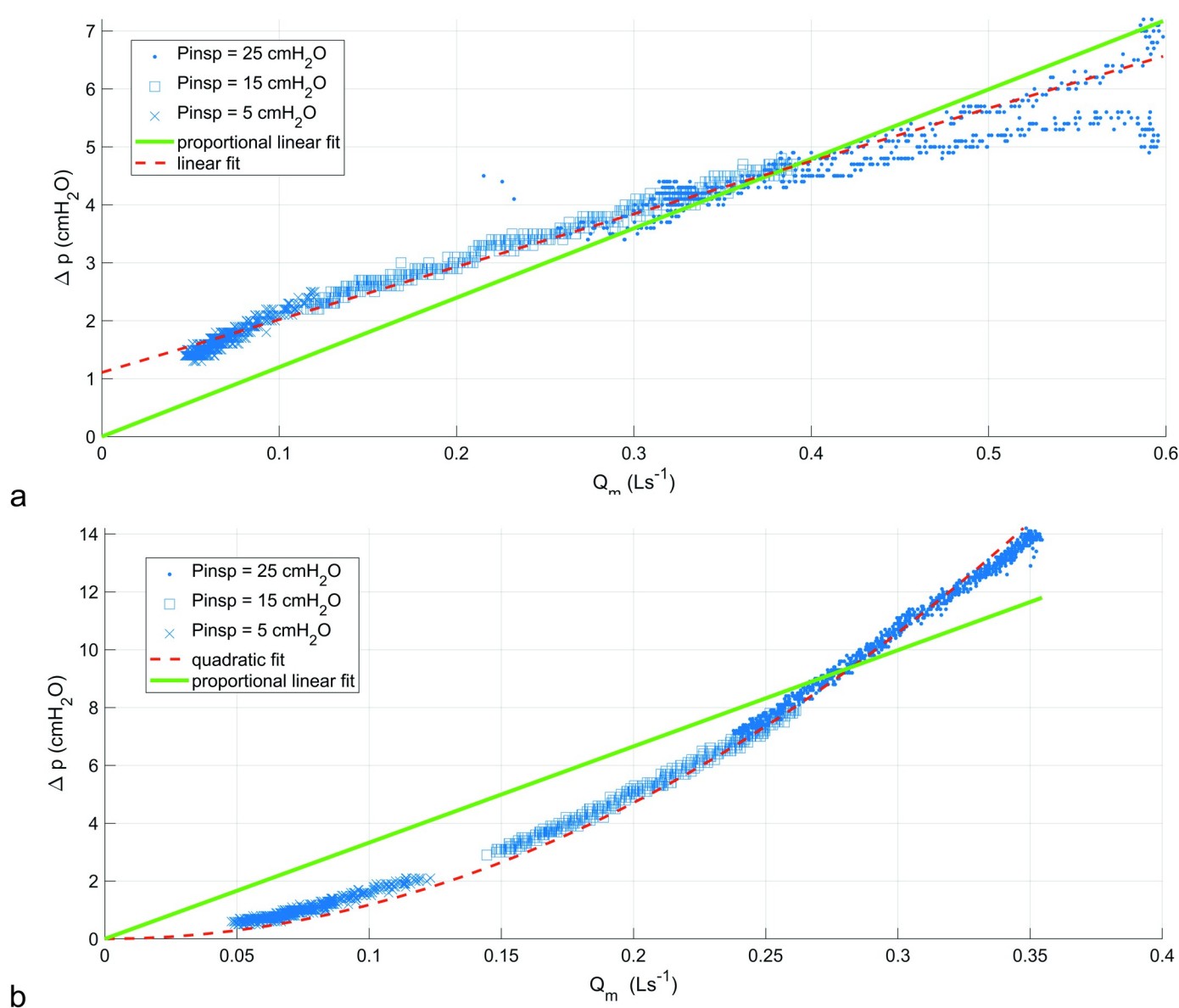

**Fig 3. Characterisation plots from testing data using Circuit 1.** a. is for the non-return valve, b is for the SOR device.

Similarly, when operating at 15 cmH$_2$O Pinsp for validation tests 4 (Fig 4D, top row) and 5 (Fig 4E, top row), the effect of adding the non-return valve to provide restriction in test 5 is clearly seen. In both cases the model predictions matched the measured data well. Lung 1 tidal

**Table 1. Fitting results for the two types of restrictor, for Pinsp values of 5, 15 and 25 cmH$_2$O, with PEEP = 5 cmH$_2$O, RR = 15 breaths/min, and I:E ratio is 1:2 for all tests.**

| Restrictor | Fit type | R$^2$ | K$_2$ (95% CI) | K$_1$ (95% CI) | K$_0$ (95% CI) |
|---|---|---|---|---|---|
| Non-return valve | unconstrained linear | 0.99 | - | 9.12 (9.07, 9.17) | 1.11 (1.09, 1.12) |
| | proportional | 0.75 | - | 11.98 (11.87, 12.1) | 0 |
| SOR device | quadratic | 0.99 | 96.84 (96.29, 97.39) | 6.07 (5.92, 6.23) | 0 |
| | proportional | 0.89 | - | 33.28 (32.99, 33.58 | 0 |

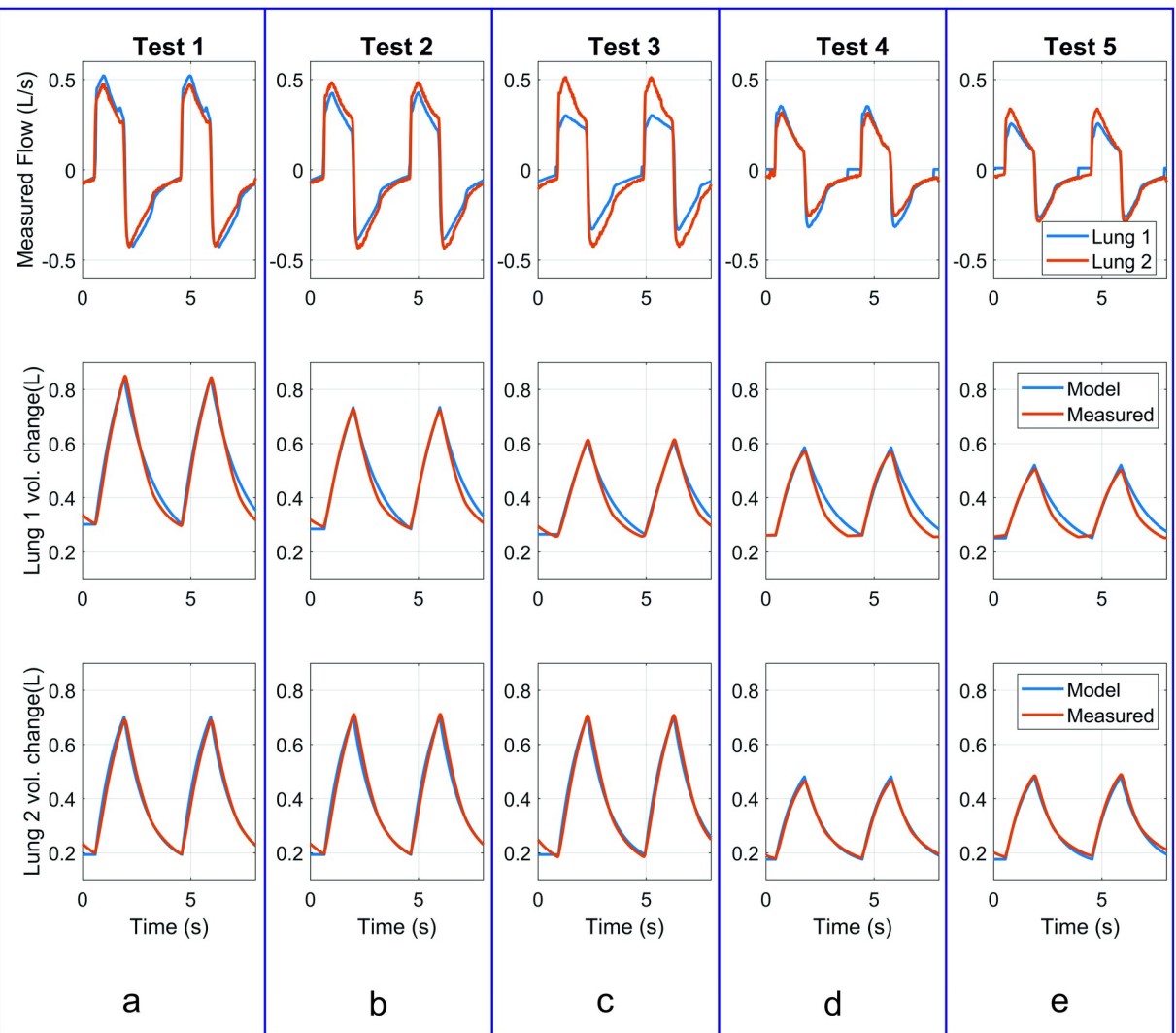

**Fig 4. Plots of measured data, together with model prediction for the validation tests made using Circuit 3.** a. Test 1, b. Test 2, c. Test 3, d. Test 4, e. Test 5.

volume reduces by 20% as a result of the restriction, again associated with a small increase in Lung 2 tidal volume (6%).

The measured and predicted values for tidal volumes are given in Table 2 for the validation tests. The RC model predicts the tidal volumes reasonably well for this range of conditions. For Lung 1, in the five validation tests the measured tidal volume deviated from the predicted by 2.4%, -3.6%, 3.5%, -5.6% and -9.3%, respectively. For Lung 2, the measured tidal volume deviated from the predicted by -2.2%, 1.2%, 2.6%, -7.5% and -2.0% respectively. The largest absolute error was 25 mL.

## Discussion

To succeed with DPV, we believe it is essential to independently control the gas flow (tidal volume that each patient receives. A possible solution is to use a flow restrictor in the line of a patient who would otherwise receive too much pressure or flow, resulting in barotrauma or

**Table 2. Test conditions, with predicted (from BathRC model) and measured tidal volumes, $V_T$.**

| Test condition | | | Tidal volume (L) | | | |
|---|---|---|---|---|---|---|
| | | | Lung 1 | | Lung 2 | |
| Validation test no. & related fig. | Pinsp (cmH$_2$0) | Restrictor (Lung 1) | Predicted | Measured (stdev*) | Predicted | Measured (stdev*) |
| 1) Fig 4A | 25 | None | 0.541 | 0.554 (0.001) | 0.509 | 0.498 (0.005) |
| 2) Fig 4B | 25 | Non-return valve[1] | 0.449 | 0.433 (0.002) | 0.509 | 0.515 (0.004) |
| 3) Fig 4C | 25 | SOR[2] | 0.345 | 0.357 (0.004) | 0.509 | 0.522 (0.001) |
| 4) Fig 4D | 15 | None | 0.324 | 0.306 (0.007) | 0.305 | 0.282 (0.006) |
| 5) Fig 4E | 15 | Non-return valve[1] | 0.270 | 0.245 (0.005) | 0.305 | 0.299 (0.006) |

[1] Resistance 12 cmH$_2$0/(L/s) [2] Resistance 33 cmH$_2$0/(L/s)

*stdev = standard deviation in measured tidal volume over 15 cycles

PEEP = 5 cmH$_2$O, RR = 15 breaths/min, and I:E ratio was 1:2 for all tests

volutrauma. Such a restrictor should ideally be adjustable. We have presented experimental results using a pair of test lungs showing that restricting the flow in one inspiration line does indeed reduce the tidal volume in the corresponding lung. Moreover, using a linear resistance-compliance network model, we have shown that the change of tidal volume can be predicted. In the five tests presented, the largest prediction error was 25 mL of tidal volume. In the form used, the model just needs an airway resistance and lung compliance estimate for each patient, which is routinely available in a clinical setting, and a resistance and compliance value for the ventilator tubing system. Likewise, an added flow restrictor should be characterised by a linear resistance i.e. a pressure drop proportional to flowrate. The predictions of the model are good despite clear linearization errors for the two flow restrictors used in this study. All the parameter values used have been informed by individual component testing.

Some further observations on the results:

1) Non-return valves are used in the individual inspiratory and expiratory limbs of each circle system (four in total). The valves we used have a considerable resistance, 12 cmH$_2$0/(L/s), so contributing over half of the total flow path resistance (either inspiration or expiration) estimated to be 22 cmH$_2$0/(L/s). While this will mean the characteristics of the dual arrangement are markedly different from conventional single patient ventilation, the increased pressure loss within the flow path means that the airway flow and pressure will be less sensitive to changes in patient characteristics.

2) A result that was not predicted by the simple modelling was that as the flow reduced to one test lung by the introduction of a flow restrictor, there was a small increase in flow to the other lung. The most severe flow restrictor reduced tidal volume by 36% in the corresponding lung, but also increased the tidal volume by 5% in the unaltered loop for the other lung.

A key advantage of the BathRC model is that it is simple to implement and does not require iterative methods. As such, it can be straightforwardly implemented in a spreadsheet (S2 File), or as an online calculator. This allows clinicians to estimate the flow restriction needed to match patient requirements. The challenge remains, however, to source a flow restrictor which is clinically acceptable, and ideally adjustable. The 3D printed designs that are emerging need to be proven to be inert, sterilisable, and durable in the breathing system environment. The fixed restrictor used in this study–in fact a non-return valve–is clinically approved and provided around 10% differentiation between the two loops. Two or more could be used in series to provide a greater restriction, but an adjustable flow restrictor would be far easier to use and would limit the need to break the closed system to add additional resistance should it be required.

Some other issues which should be investigated are:

1) The effectiveness of the non-return valves in preventing retrograde flows between patients, and in ensuring unidirectional flow around each circuit, contributing to expired $CO_2$ removal

2) The addition of sensors to give immediate feedback of the effect of flow restriction

3) The ability of a ventilator to maintain the specified pressure when the flow demands have doubled due to dual ventilation needs to be assured, as is the effectiveness of the $CO_2$ absorber, although it is anticipated that high gas flows will be used

To reiterate, the APSF have recently recommended that ventilator sharing should not be undertaken [3]. Some of the objections raised are addressed in this work. We recognise that no-one would choose to share a ventilator between two patients, but there may be some situations when there will be no choice. We also recognise the additional challenge this set-up will present to those caring for patients in these circumstances. We believe that manageability and safety mandates limiting the sharing to two patients and not more. Dual patient ventilation is a method of last resort.

## Supporting information

**S1 File.** Appendix A.
(DOCX)

**S2 File.** Appendix B.
(DOCX)

## Acknowledgments

We thank BMI Bath Clinic for providing test facilities. We also wish the acknowledge help and advice from the following University of Bath academics: Mauro Carnevale, David Cleaver, Andrew Cookson, Kate Fraser, Pejman Iravani, Evros Loukaides, Alexander Lunt, Anna Young and other members of the Bath Mechanical Ventilation Group.

## Author Contributions

**Conceptualization:** Andrew R. Plummer, Joseph M. Flynn, Malcolm Thornton, Harinderjit S. Gill.

**Data curation:** Jonathan L. du Bois.

**Formal analysis:** Andrew R. Plummer, Jonathan L. du Bois, Jens Roesner.

**Investigation:** Andrew R. Plummer, Siu Man Lee, Patrick Magee, Andrew Padkin, Harinderjit S. Gill.

**Methodology:** Andrew R. Plummer, Jonathan L. du Bois, Joseph M. Flynn, Jens Roesner, Siu Man Lee, Patrick Magee, Malcolm Thornton, Harinderjit S. Gill.

**Project administration:** Andrew R. Plummer, Harinderjit S. Gill.

**Resources:** Siu Man Lee, Patrick Magee, Malcolm Thornton, Andrew Padkin.

**Software:** Jonathan L. du Bois.

**Supervision:** Andrew R. Plummer, Patrick Magee, Andrew Padkin, Harinderjit S. Gill.

**Validation:** Jonathan L. du Bois.

**Visualization:** Jonathan L. du Bois, Harinderjit S. Gill.

**Writing – original draft:** Andrew R. Plummer.

**Writing – review & editing:** Jonathan L. du Bois, Joseph M. Flynn, Jens Roesner, Siu Man Lee, Patrick Magee, Malcolm Thornton, Andrew Padkin, Harinderjit S. Gill.

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
