## [Decision Letter · Decision Letter 0]

24 Sep 2020

PONE-D-20-16429

A Simple Method to Estimate Flow Restriction for Dual Ventilation of Dissimilar Patients: The BathRC Model

PLOS ONE

Dear Dr. Gill,

Thank you for submitting your manuscript to PLOS ONE. After careful consideration, we feel that it has merit but does not fully meet PLOS ONE’s publication criteria as it currently stands. Therefore, we ask for minor revision based on the comments, and invite you to submit a revised version of the manuscript that addresses the points raised during the review process.

We look forward to receiving your revised manuscript.

Kind regards,

Yong Wang, PhD

Academic Editor

PLOS ONE

Journal Requirements:

We note that one or more of the authors are employed by a commercial company: BMI Bath Clinic,.

2.1. Please provide an amended Funding Statement declaring this commercial affiliation, as well as a statement regarding the Role of Funders in your study. If the funding organization did not play a role in the study design, data collection and analysis, decision to publish, or preparation of the manuscript and only provided financial support in the form of authors' salaries and/or research materials, please review your statements relating to the author contributions, and ensure you have specifically and accurately indicated the role(s) that these authors had in your study. You can update author roles in the Author Contributions section of the online submission form.

2.2. Please also provide an updated Competing Interests Statement declaring this commercial affiliation along with any other relevant declarations relating to employment, consultancy, patents, products in development, or marketed products, etc.  

Reviewers' comments:

Reviewer's Responses to Questions

**Comments to the Author**

1. Is the manuscript technically sound, and do the data support the conclusions?

Reviewer #1: Yes

Reviewer #2: Yes

2. Has the statistical analysis been performed appropriately and rigorously? 

Reviewer #1: Yes

Reviewer #2: Yes

3. Have the authors made all data underlying the findings in their manuscript fully available?

Reviewer #1: Yes

Reviewer #2: Yes

4. Is the manuscript presented in an intelligible fashion and written in standard English?

Reviewer #1: Yes

Reviewer #2: Yes

5. Review Comments to the Author

Reviewer #1: The COVID-19 pandemic is a global crisis, and it has renewed the interest of Dual Patient Ventilation. The authors constructed a simple model namely BathRC model. They validated the model by using a clinical ventilator connected to two test lungs, and proved that the model can be used to estimate required flow restriction for DPV. The manuscript presents an interesting work in estimation of flow restriction for DPV. The present study must have wide interests of readers, and the paper is well prepared. Its topic is prettily fitting the aims of PLOS ONE. I recommend it to be accepted for publication after the following minor-issues to be addressed.

1. Page 7 lines 147-149, the authors stated that “For each flow restrictor, all pressure drop measurements were plotted against the mean flowrate (Q m ),…A bi-square weighted robust least squares fitting method”. Why did the authors selected such the function to fit the data? Would the authors give more details about the reason of choosing the fitting function?

2. The authors validated the new model by using two test lungs. Is there any different between test lungs and human lungs? Could the model exactly be used for the real clinical trial? Please explain it, which does not weaken merits of the new model.

Reviewer #2: The authors have done a much more in-depth experimental study than previous papers on this topic. They have also derived a very simple model and non-iterative equation that helps with coming up with a very good estimate of the amount of flow resistance that should be added to ensure that each patient receives the appropriate flow.

I haven't done much experimental work on this topic, so don't really feel I can comment on that part in detail, but they seem to me to have done a sufficiently rigorous piece of work. The experimental results are in excellent agreement with the model.

I can, however, comment on the derivation in supplementary material S1.doc, since this uses familiar circuit theory that most engineers should be able to understand. The qualitative structure of the final equation (22) is correct. However, I am a little bit uneasy with the way the derivation ignored the 2nd order term going from (3) to (4) in an arguably hand-wavy manner- there is no accompanying quantitative analysis done to justify this or citation to a paper where this was done. I would have preferred a derivation which would just state upfront that a 1st order model will be assumed, because of experimental observations, and then validate the 1st order model, which is in pretty good agreement anyway. I suspect one would get a very similar-looking equation as in (22), with slight differences in notation. However, all quantitative results and conclusions will still be the same. I therefore strongly encourage the authors to consider this in their revision.

Related questions that could have been explored, but weren't, are

(i) whether the second order model is always over-damped and

(ii) what the possible range of errors could be if one were to approximate the 2nd order dynamics with a 1st order model.

However, this is probably the topic for another paper - I wouldn't be surprised if this has already been done elsewhere in the circuit theory or lung modelling literature? Perhaps they would like to comment on this in their revision?

6. PLOS authors have the option to publish the peer review history of their article (what does this mean?). If published, this will include your full peer review and any attached files.

Reviewer #1: No

Reviewer #2: No

---

## [Author Response · Author response to Decision Letter 0]

7 Oct 2020

Editorial Comments

 RESPONSE: changes made as requested

We note that one or more of the authors are employed by a commercial company: BMI Bath Clinic,.

2.1. Please provide an amended Funding Statement declaring this commercial affiliation, as well as a statement regarding the Role of Funders in your study. If the funding organization did not play a role in the study design, data collection and analysis, decision to publish, or preparation of the manuscript and only provided financial support in the form of authors' salaries and/or research materials, please review your statements relating to the author contributions, and ensure you have specifically and accurately indicated the role(s) that these authors had in your study. You can update author roles in the Author Contributions section of the online submission form.

2.2. Please also provide an updated Competing Interests Statement declaring this commercial affiliation along with any other relevant declarations relating to employment, consultancy, patents, products in development, or marketed products, etc. 

 RESPONSE: The requested changes have been made.

RESPONSE: The DOI has now been assigned, it is 10.15125/BATH-00816, this has been added at the appropriate point.

Reviewers' comments:

Reviewer #1: The COVID-19 pandemic is a global crisis, and it has renewed the interest of Dual Patient Ventilation. The authors constructed a simple model namely BathRC model. They validated the model by using a clinical ventilator connected to two test lungs, and proved that the model can be used to estimate required flow restriction for DPV. The manuscript presents an interesting work in estimation of flow restriction for DPV. The present study must have wide interests of readers, and the paper is well prepared. Its topic is prettily fitting the aims of PLOS ONE. I recommend it to be accepted for publication after the following minor-issues to be addressed.

1. Page 7 lines 147-149, the authors stated that “For each flow restrictor, all pressure drop measurements were plotted against the mean flowrate (Q m ),…A bi-square weighted robust least squares fitting method”. Why did the authors selected such the function to fit the data? Would the authors give more details about the reason of choosing the fitting function?

RESPONSE: The bi-square weighting fitting method was used for outlier rejection. The quadratic function represents what is seen with a combination of turbulent and laminar losses in a flow. The constant term in the quadratic is manifest when there is a spring-operated valve in the flow path.

We have now clarified the reasons for using the bi-square fitting method and for the choice of the quadratic function in the manuscript.

2. The authors validated the new model by using two test lungs. Is there any different between test lungs and human lungs? Could the model exactly be used for the real clinical trial? Please explain it, which does not weaken merits of the new model.

RESPONSE: The test lungs provide a safe and repeatable way to test the ventilation system and have compliances and volumes with the typical adult range. There will be variability between human subjects, however we believe the model should be able to predict the behaviour of human lungs.

Reviewer #2: The authors have done a much more in-depth experimental study than previous papers on this topic. They have also derived a very simple model and non-iterative equation that helps with coming up with a very good estimate of the amount of flow resistance that should be added to ensure that each patient receives the appropriate flow.

I haven't done much experimental work on this topic, so don't really feel I can comment on that part in detail, but they seem to me to have done a sufficiently rigorous piece of work. The experimental results are in excellent agreement with the model.

I can, however, comment on the derivation in supplementary material S1.doc, since this uses familiar circuit theory that most engineers should be able to understand. The qualitative structure of the final equation (22) is correct. However, I am a little bit uneasy with the way the derivation ignored the 2nd order term going from (3) to (4) in an arguably hand-wavy manner- there is no accompanying quantitative analysis done to justify this or citation to a paper where this was done. I would have preferred a derivation which would just state upfront that a 1st order model will be assumed, because of experimental observations, and then validate the 1st order model, which is in pretty good agreement anyway. I suspect one would get a very similar-looking equation as in (22), with slight differences in notation. However, all quantitative results and conclusions will still be the same. I therefore strongly encourage the authors to consider this in their revision.

Related questions that could have been explored, but weren't, are

(i) whether the second order model is always over-damped and

(ii) what the possible range of errors could be if one were to approximate the 2nd order dynamics with a 1st order model.

However, this is probably the topic for another paper - I wouldn't be surprised if this has already been done elsewhere in the circuit theory or lung modelling literature? Perhaps they would like to comment on this in their revision?

RESPONSE:

Reviewer 2 asks that a justification be given in the assumption made to go from Eqn. (3) to Eqn. (4). This has been provided, using nominal parameter values from the experimental cases to give an indication of the order of magnitude of the errors introduced by this assumption. Further suggestions in (i) and (ii) are, as the reviewer says, too large a topic for inclusion in full here, but these have been addressed to some extent: for (i) we have provided the answer but not the proof; and for (ii) we have provided a quantification of nominal values but not a full study of the range of these values. This is all included in the short paragraph newly introduced after Eqn. (4). For the reviewer’s scrutiny, the full derivations are provided below.

Paragraph included after Eqn. (4)

It can be shown that the system in Eqn. 3 is always overdamped for physically possible values of R, C, R_v and C_v (i.e. positive values). The second order system is thus comprised of two first-order lags. Using the nominal parameter values from the experiments below, these lags have time constants of 1.25 s and 0.0296 s. For comparison, the first order system in Eqn. 4 has a single time constant of 1.28 s. This value is very close to one of the second order time constants, while the remaining time constant is too small to have a significant contribution at the timescales considered. Transfer functions for the first-order system and the second-order system differ by less than 2% in amplitude at the nominal respiration rate of 0.25 Hz.

Damping in second order system

The second order system in eqn. 3 is always over-damped, and this is proven as follows. From eqn. 3, the damping ratio for the system is

ζ=(RC+R_v C_v+R_v C)/(2√(RCR_v C_v ))

Substituting ϕ_R=R_v/R and ϕ_C=C_v/C gives

ζ=(1+ϕ_R ϕ_C+ϕ_R)/(2√(ϕ_R ϕ_C )).

For an over-damped response, ζ-1>0, or

(1+ϕ_R ϕ_C+ϕ_R-2√(ϕ_R ϕ_C ))/(2√(ϕ_R ϕ_C ))>0.

For positive, real values of ϕ_R and ϕ_C (i.e. physically possible values) the denominator can be eliminated to give the criterion for an over-damped response as

1+ϕ_R ϕ_C+ϕ_R-2√(ϕ_R ϕ_C )>0

or equivalently

(1+ϕ_R ϕ_C+ϕ_R )^2-4ϕ_R ϕ_C>0.

Defining

f(ϕ_R,ϕ_C )=(1+ϕ_R ϕ_C+ϕ_R )^2-4ϕ_R ϕ_C

then setting ϕ_R and ϕ_C to zero gives f(ϕ_R,ϕ_C )=1, a positive result. So if an under-damped configuration exists for positive ϕ_R and ϕ_C then the function f(ϕ_R,ϕ_C ) would need to transition from positive to negative and there would need to be a solution that satisfies f(ϕ_R,ϕ_C )=0, or

1+2ϕ_R+〖ϕ_R〗^2 〖ϕ_C〗^2+2〖ϕ_R〗^2 ϕ_c+〖ϕ_R〗^2-2ϕ_R ϕ_C=0.

Rearranging gives

〖ϕ_R〗^2 (〖ϕ_C〗^2+2ϕ_c+1)+ϕ_R (2-2ϕ_C )+1=0

and solving for ϕ_R gives

ϕ_R=(ϕ_C-1±√(-4ϕ_c ))/(〖ϕ_C〗^2+2ϕ_c+1).

A real valued solution for ϕ_R only exists for non-positive values of ϕ_C, which are physically impossible, so it is concluded that ζ-1>0 for all physically possible values of R, C, R_v and C_v. The second order system in eqn. 3 will therefore always be overdamped.

Approximation to first order system

The second order system in eqn. 3 has the characteristic equation

〖RCR_v C_v s〗^2+(RC+R_v C_v+R_v C)s+1=0

which yields roots at

s=-(RC+R_v C_v+R_v C)/(2RCR_v C_v )±√(((RC+R_v C_v+R_v C)/(2RCR_v C_v ))^2-1/(RCR_v C_v )).

The system is over-damped so will have two real roots corresponding to the time constants of two first-order lags. Neglecting the second order term, the root of the resulting first order system is

s=(-1)/(RC+R_v C_v+R_v C).

The first order system is a good approximation to the second order system if: (a) one of the two second order poles is close to the first order pole, (b) there is good separation between the two second order poles, and (c) the operating frequency is well below that of the highest second-order pole.

Using the nominal values from the experiment in the first row of Table 2, the time constants for the first order system are 0.0296 s and 1.25 s, and the time constant for the second order system is 1.28 s. The first order system matches the slowest mode of the second order system to within 2.4%, satisfying criterion (a). The two modes of the second order system are separated by a factor of 42, satisfying criterion (b). The operating frequency is 15 breaths/min or 0.25 Hz, which is well below the 5.4 Hz of the fastest second order mode, satisfying criterion (c).

At the operating frequency of 15 breaths/min the amplitude and phase response of the second order system is -1.62×10-7 m3/Pa and -65.6o while that of the first order system is -1.59×10-7 m3/Pa and -63.5o, giving an amplitude difference of less than 2% and a phase difference of only 2.1o. The first order system thus provides a good approximation to the second order system in the context of this study.

---

## [Decision Letter · Decision Letter 1]

28 Oct 2020

A Simple Method to Estimate Flow Restriction for Dual Ventilation of Dissimilar Patients: The BathRC Model

PONE-D-20-16429R1

Dear Dr. Gill,

We’re pleased to inform you that your manuscript has been judged scientifically suitable for publication and will be formally accepted for publication once it meets all outstanding technical requirements.

Kind regards,

Yong Wang, PhD

Academic Editor

PLOS ONE

Additional Editor Comments (optional):

Reviewers' comments:

Reviewer's Responses to Questions

**Comments to the Author**

1. If the authors have adequately addressed your comments raised in a previous round of review and you feel that this manuscript is now acceptable for publication, you may indicate that here to bypass the “Comments to the Author” section, enter your conflict of interest statement in the “Confidential to Editor” section, and submit your "Accept" recommendation.

Reviewer #1: All comments have been addressed

Reviewer #2: All comments have been addressed

2. Is the manuscript technically sound, and do the data support the conclusions?

Reviewer #1: Yes

Reviewer #2: Yes

3. Has the statistical analysis been performed appropriately and rigorously? 

Reviewer #1: Yes

Reviewer #2: Yes

4. Have the authors made all data underlying the findings in their manuscript fully available?

Reviewer #1: Yes

Reviewer #2: Yes

5. Is the manuscript presented in an intelligible fashion and written in standard English?

Reviewer #1: Yes

Reviewer #2: Yes

6. Review Comments to the Author

Reviewer #1: (No Response)

Reviewer #2: No further comments. The reviewers have adequately addressed all my comments from the previous review.

7. PLOS authors have the option to publish the peer review history of their article (what does this mean?). If published, this will include your full peer review and any attached files.

Reviewer #1: No

Reviewer #2: No

---

## [Editor Report · Acceptance letter]

3 Nov 2020

PONE-D-20-16429R1 

A Simple Method to Estimate Flow Restriction for Dual Ventilation of Dissimilar Patients: The BathRC Model 

Dear Dr. Gill:

I'm pleased to inform you that your manuscript has been deemed suitable for publication in PLOS ONE. Congratulations! Your manuscript is now with our production department. 

Kind regards, 

on behalf of

Dr. Yong Wang 

Academic Editor

PLOS ONE